# Role of ERβ in Triple-Negative Breast Cancer Associated with p53 and Androgen Receptor

**DOI:** 10.3390/ijms262311459

**Published:** 2025-11-26

**Authors:** Kei Ito, Naoko Honma, Hideaki Ogata, Akimitsu Yamada, Mika Miyashita, Tomio Arai, Eiichi Sasaki, Kazutoshi Shibuya, Tetuo Mikami, Masataka Sawaki

**Affiliations:** 1Department of Pathology, Toho University Faculty of Medicine, Omori-Nishi 5-21-16, Ota-ku, Tokyo 143-8540, Japan; k-ito@tius.ac.jp (K.I.); tetsuo.mikami@med.toho-u.ac.jp (T.M.); 2Department of Medical Technology, Faculty of Health Sciences, Tsukuba International University, Manabe 6-20-1, Tsuchiura 300-0051, Japan; 3Department of Surgical Pathology, Toho University Faculty of Medicine, Omori-Nishi 5-21-16, Ota-ku, Tokyo 143-8540, Japan; kaz@med.toho-u.ac.jp; 4Department of Breast and Endocrine Surgery, Toho University Omori Medical Center, Omori-Nishi 6-11-1, Ota-ku, Tokyo 143-8541, Japan; ogatah@med.toho-u.ac.jp; 5Department of Gastroenterological Surgery, Yokohama City University Graduate School of Medicine, Fukuura 3-9, Kanazawa-ku, Yokohama 236-0004, Japan; yakimitsu@gmail.com; 6Palliative Care Nursing, Department of Human Health Sciences, Graduate School of Medicine, Kyoto University, Shogoin-Kawahara-cho 53, Sakyo-ku, Kyoto 606-8507, Japan; miyashita.mika.3w@kyoto-u.ac.jp; 7Department of Pathology, Tokyo Metropolitan Institute for Geriatrics and Gerontology, Sakaecho 35-2, Itabashi-ku, Tokyo 173-0015, Japan; arai@tmghig.jp; 8Department of Pathology and Molecular Diagnostics, Aichi Cancer Center Hospital, Kanokoden 1-1, Chikusa-ku, Nagoya 464-8681, Japan; esasaki@aichi-cc.jp; 9Department of Breast Oncology, Nagoya Medical Center, National Hospital Organization 4-1-1 Sannomaru, Naka-ku, Nagoya 460-0001, Japan; masataka.sawaki@outlook.jp

**Keywords:** androgen receptor, clinical outcome, estrogen receptor-β, p53, triple-negative breast cancer

## Abstract

In triple-negative breast cancer (TNBC), the clinicopathological significance of the expression of a second estrogen receptor, ERβ, remains unclear. Further, although the clinicopathological significance of mutant p53 and androgen receptor (AR) has been investigated in TNBC, they have not been established as therapeutic targets. Experimental studies reported the importance of cross-talk between ERβ and p53 or AR in TNBC. In this study, we immunohistochemically examined ERβ expression in surgical specimens of TNBC obtained from postmenopausal patients who underwent surgery without neoadjuvant therapy and investigated the relationship between ERβ expression and various clinicopathological factors, including clinical outcome, while also considering p53 and AR. No significant difference in clinical outcome was noted according to the ERβ status alone (*p* = 0.2908). However, the ERβ status did affect the relationship between the clinical outcome and p53 or AR status; p53-positive or AR-positive group exhibited significantly more favorable clinical outcomes than p53-negative or AR-negative group, respectively, in the ERβ-positive group (p53, *p* = 0.0265; AR, *p* = 0.0285), but not in the ERβ-negative group (p53, *p* = 0.7228; AR, *p* = 0.7734). This may be the result of a functional interaction between ERβ and p53 or AR. The role of ERβ in TNBC will be elucidated in further complex studies considering multiple molecules.

## 1. Introduction

Since triple-negative breast cancer (TNBC) lacks an effective treatment target, that is, classic estrogen receptor (ERα), progesterone receptor (PgR), or human epidermal growth factor receptor-2 (HER2), chemotherapy is the most established pharmacotherapy for patients with TNBC. Most cases of TNBC involve an undifferentiated tumor and are biologically aggressive, resulting in unfavorable outcomes for patients [1]. TNBC is relatively frequent in younger women or specific ethnicities and is closely related to germline gene mutations, such as BRCA, in these patients. The therapeutic options available for patients with TNBC have been increasing, such as PARP inhibitors for BRCA-mutated patients, immune checkpoint inhibitors for PD-L1-positive tumors [1], or oncolytic peptides in the context of cancer immunotherapy [2]. TNBC, however, is known to be heterogeneous, suggesting the need for appropriate biomarkers to optimize treatment. It may be meaningful to focus on TNBC in postmenopausal women, including older women, who may benefit from pharmacotherapies other than the current options. Although TNBC is characterized as non-responsive to estrogen and its related systems, the expression of receptors for sex steroid hormones other than ERα has now been demonstrated in TNBC, and the molecular mechanisms and clinical relevance of these receptors have been investigated [3,4,5,6,7,8,9,10,11,12,13]. Receptors for estrogen, including a second ER (ERβ) or G protein-coupled estrogen receptor (GPER), and that for androgen (AR), are known, but their roles in TNBC remain unclear. From a clinicopathological point of view, AR has been reported to be expressed in about 40% of TNBC cases, forming the so-called ‘luminal AR’ subtype (LAR), for which AR-targeted therapy has been in a clinical trial [13]. The clinicopathological role of AR has been examined in many studies, but its localization, antibodies, and cut-off values vary, with some reports suggesting a favorable prognostic effect of AR, some suggesting a poor prognostic effect, and others suggesting no prognostic role at all [14,15,16,17,18,19]. We recently showed that AR expression is closely related to patient age [18]. ERβ has also been reported to have various effects in TNBC, which at least partly reflect the differences in study methodologies (in vitro or in vivo experimental studies, real-time RT-PCR or immunohistochemistry on surgical materials, antibody used, intracellular structure estimation, cut-off values, etc.) or settings (eligibility of included patients). Honma et al. reported favorable clinical outcomes in a group of postmenopausal ERβ-positive TNBC patients treated with adjuvant tamoxifen monotherapy [20], whereas others reported an unfavorable role of ERβ [21,22,23]. The role of p53 in TNBC is also controversial [24]. p53 is a representative cancer suppressor protein, inducing cell cycle arrest or apoptosis in genetically damaged cells [25,26,27]. Aberrant p53 promotes tumors [25] and is reportedly frequent in TNBC. Tumors with mutated *TP53* have been reported to frequently achieve pathologic complete remission with neoadjuvant chemotherapy [28]. In addition to the respective roles of ERβ, AR, and p53 in TNBC, there have also been reports of molecular cross-talk between ERβ and p53 or AR in basic studies [12,29,30,31,32,33]. In this study, we examined the clinicopathological role of ERβ alone or in combination with p53 and AR, based on immunohistochemical findings.

## 2. Results

Typical immunostaining images of ERβ, p53, and AR are shown in Figure 1. Of 122 patients, 91 (75%), 90 (74%), and 67 (55%) were positive for ERβ, p53, and AR, respectively. The relationship between ERβ expression and various clinicopathological factors, including patients’ age, tumor size, nodal status, histological grade, Ki-67 status, AR status, and p53 status, were examined; however, no significant correlation was observed, except for Ki-67 status, which showed a positive correlation with ERβ expression (*p* = 0.009, Table 1). Although insignificant, ERβ positivity was related to smaller tumor size (*p* = 0.128), negative nodal metastasis (*p* = 0.142), and lower pStage (*p* = 0.060) (Table 1). Aside from ERβ, AR and p53 positively correlated with each other (*p* = 0.0211, Table 2). Regarding the clinical outcome, we focused on the disease-free interval (DFI), in which recurrence/metastasis was considered as event, to elucidate the outcome of the tumor itself, specifically in older patients, who frequently die from causes other than the tumor. No significant difference in DFI was observed according to ERβ status alone, though positivity was slightly related to a favorable outcome in total (*p* = 0.2908) and in the older group (*p* = 0.2063), though this was not observed in the younger group (*p* = 0.8391, Figure 2). As for p53 in total, DFI in the p53-positive group was almost significantly more favorable than in the p53-negative group (*p* = 0.0571), and this was clearer in the ERβ-positive group yielding significance (*p* = 0.0265), contrasting with the ERβ-negative group where the difference in DFI according to p53 status disappeared (*p* = 0.7228, Figure 3). Similar phenomena were observed for AR. A marginally significant difference in DFI according to AR status was shown in total (*p* = 0.0468, more favorable DFI for AR-positive group than AR-negative group); however, the difference in DFI between AR-positive and -negative groups was even clearer in the ERβ-positive group (*p* = 0.0285) but was lost in the ERβ-negative group (*p* = 0.7734, Figure 4). Since ERβ was positively correlated with Ki-67, we conducted the same analyses, comparing DFI according to Ki-67 status in total and ERβ-positive and -negative groups; however, no significant results were obtained (*p* = 0.9447, *p* = 0.5031, and *p* = 0.7734 in total, ERβ-positive, and ERβ-negative groups, respectively; Figure 5).

## 3. Discussion

In the present study, ERβ alone had no prognostic impact on TNBC. In ERα-positive tumors, ERβ reportedly suppresses cell growth, forming a heterodimer with ERα and inhibiting its tumor-promoting effect or suppressing the transcription of the c-myc, cyclin D1, or cyclin A genes. In ERα-negative tumors, on the contrary, ERβ slightly promotes tumor growth, forming an ERβ homodimer and sending proliferative signals [13,34,35]. The positive correlation between ERβ and Ki-67 observed in this study might reflect the tumor-promoting role of ERβ in TNBC. Honma et al. [20] reported a favorable clinical outcome in an ERβ-positive TNBC patient group treated with adjuvant tamoxifen monotherapy, specifically comprising postmenopausal women (most patients were younger than 75 y/o); however, it was unclear whether the results reflected a favorable prognostic effect of ERβ independent of endocrine therapy or the predictive role of ERβ for tamoxifen therapy. Since tamoxifen was not administered to the current patients in this study, the previous results may have reflected the predictive value of ERβ for tamoxifen treatment, but not an independent prognostic value of ERβ by itself.

However, as there have been reports of molecular cross-talk between ERβ and p53 or AR [12,29,30,31,32,33], we examined the clinicopathological significance of ERβ in combination with p53 or AR and found a greater prognostic impact of p53 and AR in ERβ-positive compared with ERβ-negative patients.

In breast cancer, p53 needs to be considered by subtype due to the influence of ERα and other factors [36]. In ERα-positive cancer, estrogen signaling has been reported to suppress p53 activity, which inhibits apoptosis induced by chemotherapy and causes resistance [28], suggesting the importance of estrogen signaling in the pathobiology of p53. p53-mutated breast cancer reportedly shows a high rate of pathologic complete response with neoadjuvant chemotherapy [28]. In our previous study on p53 in TNBC, there was a trend, though insignificant, toward a better prognosis in the p53-positive group [24]. In the present study, p53 was correlated with a significantly better prognosis when limited to the ERβ-positive group. This is consistent with a basic study reporting that ERβ regulates mutant p53 and reduces invasiveness in TNBC (Figure 6) [29].

The pathobiological role of androgen is also complex in breast tumors. AR is frequently expressed in normal breast epithelial cells and is considered one of the markers of mammary epithelial cell differentiation and a favorable prognostic factor in breast cancer patients. AR in the ERα-positive state exerts an anti-estrogenic, growth-inhibitory effect and antagonizes ERα signaling [37,38], while in TNBC, AR reportedly promotes the proliferation of tumor cells or metastasis [39]; thus, the role of AR in TNBC is similar to that of ERα in ERα-positive tumors, having both favorable prognostic value and a tumor-promoting effect. With such a complicated background, the prognostic role of AR alone may be controversial in TNBC. Androgens may exhibit diverse effects depending on their interactions with other steroid receptors and related signaling pathways and have also been reported to increase ERβ gene expression irrespective of ERα expression [40]. ERβ reportedly forms a heterodimer with AR, resulting in a better prognosis by blocking AR proliferative signals in TNBC [33] (Figure 6). AR alone has a favorable prognostic value in TNBC [18]; however, in the present study, the difference in DFI according to AR status was even more marked in ERβ-positive group but disappeared in ERβ-negative group, suggesting the importance of the interaction between ERβ and AR. Furthermore, the co-expression of ERβ and AR has been reported to increase the efficacy of antihormonal therapy in TNBC [41], so the clinical significance of AR may need to be considered together with ERβ.

In the present study, expressions of AR and p53 were positively correlated. Although the biological mechanism is unknown, some molecules may mediate between them. For example, KLLN has been identified as an AR-induced tumor suppressor, and studies have shown that KLLN directly promotes p53 expression in breast cancer, causing apoptosis and cell cycle arrest [42]. Identifying such mediating molecules is necessary to elucidate the association between AR and p53.

**Figure 6 ijms-26-11459-f006:**
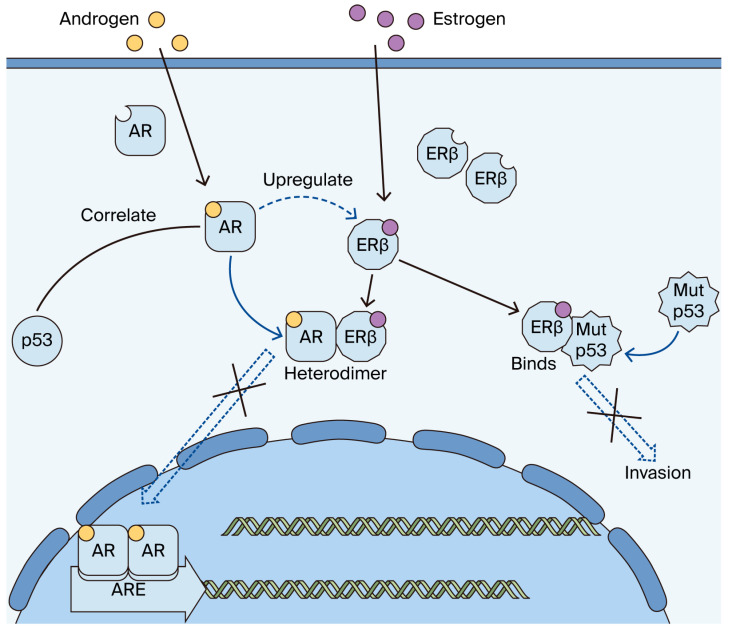
Molecular interactions between ERβ, p53, and AR in triple-negative breast cancer (TNBC) hypothesized in the present study. ERβ regulates mutant p53 and reduces invasiveness in TNBC [29]. Androgens have also been reported to increase ERβ gene expression irrespective of ERα expression [40]. ERβ reportedly forms a heterodimer with AR, resulting in a better prognosis by blocking AR proliferative signals in TNBC [33]. The co-expression of ERβ and AR has been reported to increase the efficacy of antihormonal therapy in TNBC [41]. X means blocked reactions, full arrows mean action of molecules, dashed arrow means influence of a molecule on another molecule.

Previously, in a combined investigation of Bcl-2 and p53 in TNBC, we demonstrated that a Bcl-2-negative/p53-positive group had a markedly longer DFI than the other groups [24]. These findings provide further evidence supporting the importance of combined investigations of multiple molecules and their interactions. As there has been no consistent research data regarding molecular interactions including ERβ, p53, and AR, it will be necessary to conduct further research based on a combinatory viewpoint, similar to that in this study.

The limitation of this study is its purely observational nature, as it was conducted on surgical samples using immunohistochemistry. Extensive molecular analyses are needed to prove the hypothesis presented in this study.

## 4. Materials and Methods

### 4.1. Subjects

A retrospective cross-sectional study was conducted, involving postmenopausal patients with TNBC who were surgically treated without neoadjuvant therapy at Toho University Medical Center Omori Hospital (Tokyo, Japan), Aichi Cancer Center (Nagoya, Japan), Tokyo Metropolitan Geriatric Hospital (Tokyo, Japan), Yokohama City University Hospital (Yokohama, Japan), or Kagawa University Hospital (Kagawa, Japan) in 2004–2013. Patients matched for pathological stage (pStage) and aged 75 years or older (older group, *n* = 75) and 55–64 years (younger group, *n* = 47) were included in this study as part of a consecutive series. The inclusion criteria were as follows: (1) primary invasive breast cancer; (2) TNBC, defined as <1% staining for estrogen receptor and progesterone receptor, and 0 or 1+ as per immunohistochemistry or negative as per in situ hybridization for human epidermal growth factor receptor 2 (HER2); (3) unilateral; and (4) female sex. The exclusion criteria were as follows: (1) noninfiltrating carcinoma or carcinoma with microinvasion, (2) no residual carcinoma after biopsy, (3) stage IV tumors, (4) bilateral, (5) male sex, and (6) undergoing neoadjuvant therapy.

### 4.2. Immunohistochemistry

The expression of each protein was examined immunohistochemically on representative slides of formalin-fixed and paraffin-embedded tissue, according to the established method [18,24,43,44]. Endogenous peroxidase was blocked with 0.5% hydrogen peroxide in methanol for 30 min. For antigen retrieval, the sections were treated with 98 °C Target Retrieval Solution pH 6 (for p53) or pH 9 (for ERβ and AR) (Dako, Carpinteria, CA, USA) for 40 min each. After the application of primary antibodies for ERβ (clone PPG5/10, Bio-Rad, Hercules, CA, USA) and AR (clone AR27, Novocastra, Melbourne, Australia), and p53 (clone DO-7, Dako. X1000) for 60 min at room temperature, the slides were washed with phosphate-buffered saline and incubated with secondary EnVision+System-HRP labeled polymer (Dako). Finally, tissue sections were visualized with Stable DAB (Falma, Tokyo, Japan). Nuclear immunoreactivity was estimated independently by two researchers (KI and NH), who scored both the intensity and percentage of cancer cells (Allred score). The cut-off was set at 10% for ERβ and AR. For p53, a positive result was defined as one in which tumor cells stained more strongly than adjacent noncancerous epithelial cells [18,20,24].

### 4.3. Statistical Analysis

The expression of ERβ was compared with classical clinicopathological factors and AR or p53 expression. The Chi-square test or Fisher’s exact test were used to compare categorical variables, while the disease-free interval (DFI) was analyzed using the Kaplan–Meier method and log-rank test. Data were analyzed with JMP version 13, and *p* < 0.05 was considered significant.

## Figures and Tables

**Figure 1 ijms-26-11459-f001:**
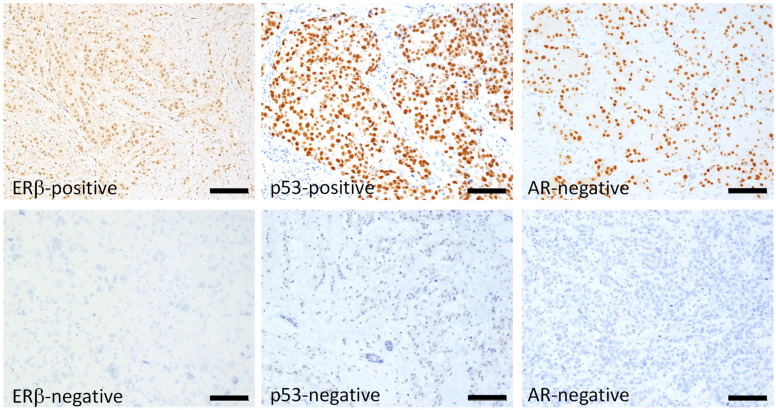
Typical positive and negative nuclear staining of breast cancerous cells for ERβ, p53, and AR: immunoperoxidase staining with hematoxylin counterstaining. Bar: 100 μm. The cut-off was set at 10% for ERβ and AR. For p53, a positive result was defined as one in which tumor cells stained more strongly than adjacent noncancerous epithelial cells.

**Figure 2 ijms-26-11459-f002:**
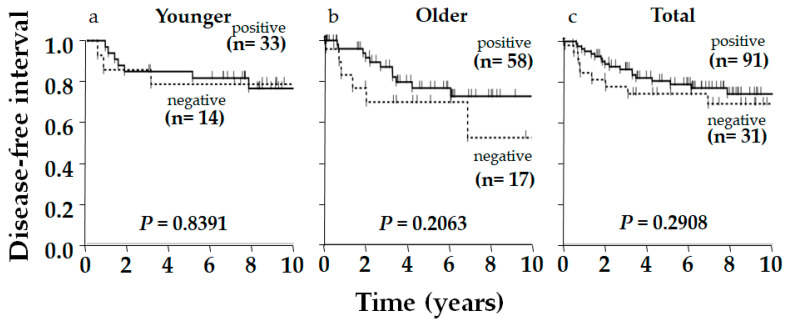
Kaplan–Meier disease-free interval (DFI) curve of postmenopausal women with triple-negative breast cancer according to estrogen receptor-β (ERβ) status. Younger: 47 patients aged 55–64 y/o; older: 75 patients aged 75 y/o or older; total: 122 postmenopausal patients. ERβ positivity was slightly related to favorable DFI in total and older patients, but the difference was insignificant. *p*-value was determined using the log-rank test.

**Figure 3 ijms-26-11459-f003:**
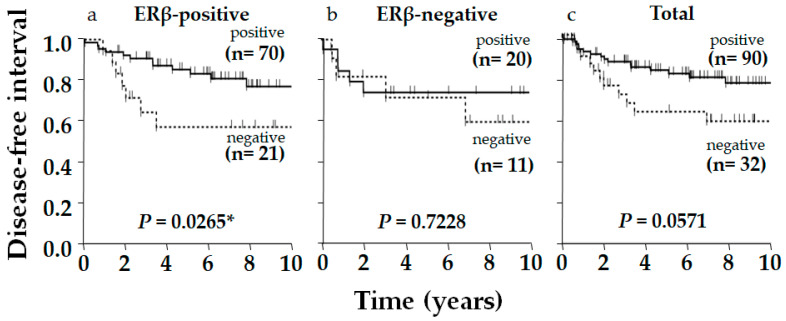
Kaplan–Meier disease-free interval (DFI) of postmenopausal women with triple-negative breast cancer according to p53 status in ERβ-positive group, ERβ-negative group, and total. In total, DFI of p53-positive group was almost significantly more favorable than p53-negative group (*p* = 0.0571), which was cleared in ERβ-positive group yielding significance (*p* = 0.0265), contrasting with ERβ-negative group where difference disappeared (*p* = 0.7228). *p*-value was determined using log-rank test. (*) Significance, *p* < 0.05.

**Figure 4 ijms-26-11459-f004:**
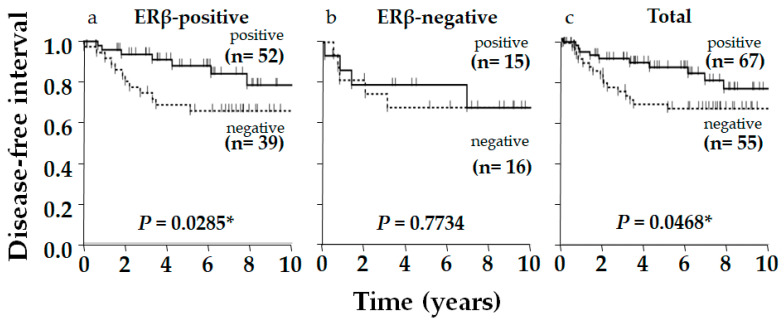
Kaplan–Meier disease-free interval (DFI) curve of postmenopausal women with triple-negative breast cancer according to androgen receptor (AR) status in ERβ-positive group, ERβ-negative group, and total. AR-positive group exhibited more favorable DFI than AR-negative group in total with marginal significance (*p* = 0.0468), which was much clearer in ERβ-positive group (*p* = 0.0285), contrasting with ERβ-negative group where difference in DFI according to AR status was lost (*p* = 0.7734). *p*-value was determined using log-rank test. (*) Significance, *p* < 0.05.

**Figure 5 ijms-26-11459-f005:**
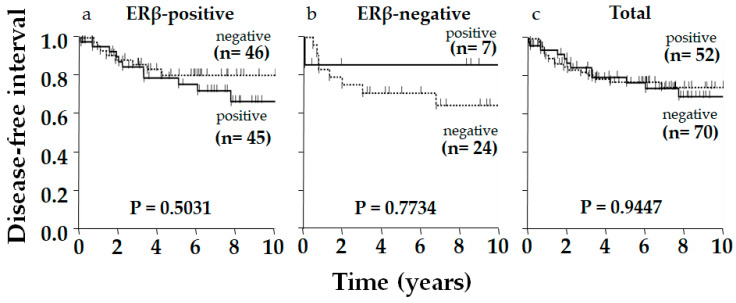
Kaplan–Meier disease-free interval (DFI) curve of postmenopausal women with triple-negative breast cancer according to Ki-67 status in the ERβ-positive group, ERβ-negative group, and total. Ki-67 status affected DFI in none of the groups. *p*-value was determined using the log-rank test.

**Table 1 ijms-26-11459-t001:** The relationship between ERβ and various clinicopathological factors. No significant correlation was observed except for Ki-67 status, which showed a positive correlation with ERβ expression.

Factors		ERβ				
		Positive	Negative	Pos %	*p*-Value ^a^	
Age (y)	Younger	33	14	70	0.38	NS
	Older	58	17	77		
Size (mm)	≤20	37	8	82	0.128	NS
	20<	53	23	70		
Nodal stats	Positive	31	16	66	0.142	NS
	Negative	54	15	78		
pStage (I and II vs. III)	III	15	10	60	0.060	NS
	I and II	76	21	78		
Histological grade	III	69	21	77	0.377	NS
	I and II	22	10	69		
Ki-67	Positive	45	7	87	0.009 *	Positive Correlation
	Negative	46	24	66		
AR	Positive	52	15	78	0.396	NS
	Negative	39	16	71		
p53	Positive	70	20	78	0.175	NS
	Negative	21	11	66		

^a^ Pearson Chi-squared test; Pos %, % of positive cases; NS, not significant; pStage, pathological stage; AR, androgen receptor. (*) Statistically significant.

**Table 2 ijms-26-11459-t002:** The relationship between the immunohistochemical statuses of AR and p53, which were positively correlated.

	p53				
	Positive	Negative	Pos %	*p*-Value ^a^	
AR-Positive	55	12	82	0.0211 *	Positive Correlation
AR-Negative	35	20	64		

^a^ Pearson Chi-squared test; Pos %, % of positive cases; (*) statistically significant.

## Data Availability

The datasets presented in this article are not readily available because the data are part of an ongoing study. Requests to access the datasets should be directed to Naoko Honma.

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
