# Peer review of "Role of ERβ in Triple-Negative Breast Cancer Associated with p53 and Androgen Receptor"

_ijms, 2025, doi:10.3390/ijms262311459_

Round 1

Reviewer 1 Report

Comments and Suggestions for Authors

Summary:

This study investigates the role of ERβ in TNBC using samples from 122 patients surgically treated without neoadjuvant therapy, categorized by age, tumor size, nodal status, Ki-67, AR, and p53 expression levels, to examine the correlations with ERβ. The results showed that ERβ alone is not a prognostic marker. However, AR and P53 status appear to influence disease-free survival in ERβ-positive patients, providing a prognostic value of P53 and AR in this group.

Overall, the manuscript addresses the correlation of AR and P53 with ERβ-positive patients. However, it lacks a sufficient background and rationale in the introduction section. The results section would benefit from a more detailed description. Additionally, the experimental design could be strengthened through a more comprehensive data analysis. Better formatting of both figure legends and references is also recommended. A key limitation of the study is its purely observational nature, analyzing correlations among clinicopathological factors in human patient samples. Therefore, the proposed mechanism link cannot be conclusively established. Detailed suggestions are provided below

Major revisions:

  1. The introduction is overly brief. Elaborating on the functions of AR, ER-β, and p53, as well as providing clear rationales for examining their respective and correlated roles in TNBC, would strengthen the background and improve the overall context. Although some of these rationales are mentioned in the discussion section, I recommend rearranging the content to improve the clarity and flow.
  2. The results section is insufficiently detailed. Figures or tables are not fully explained in the text.
  3. Figure legends should be properly formatted, including a concise title, a clear description of the staining, scale bars, statistical details, the scoring methods used, etc.
  4. Representative images of both positive and negative P53, AR and ERβ immunochemistry stainings in Fig.1 will improve the Figure quality and clarity.
  5. In Table 1, the author shows that Ki-67 positively correlates with the positivity of ERβ. What is the disease-free interval (DFI) stratified by Ki-67 in relation to ERβ? This will provide a potential prognostic value of ERβ.
  6. In the results section, lines 72 - 73, a missing figure.
  7. In the discussion, the authors propose that KLLN may directly promote p53 activity in breast cancer. However, the current evidence based on the correlation between AR and p53 is insufficient to support a link with KLLN. Performing KLLN staining on patient samples could provide additional insight, at least to assess its correlation with p53/AR.

 Minor revisions:

  1. Is there any specific reason for choosing postmenopausal women after surgery without adjuvant chemotherapy?
  2. H-score can be incorporated to better stratify the staining intensity, as shown in Fig.1 where the intensity of ERβ varies.
  3. In Table 1, the figure legend indicates both the Pearson chi-squared test and Fisher’s exact test were performed, but only the Pearson chi-squared test was reported in the table.
  4. Lines 97-100: Missing reference.
  5. Lines 140-141: include the Figure number in parentheses.
  6. The sentence from lines 150 - 155 is overly long, and the transition to discussing mTOR could be smoother.
  7. Lines 183-185, citation needed.
  8. Abbreviations section needs adjustment, for example, HER2.
  9. The Reference section needs adjustment, for example, reference 2; unknown characters like O' were included.

Comments on the Quality of English Language

The English could be improved to avoid repetition and enhance clarity. Reorganization of the content is highly recommended.

Author Response

To Reviewer 1

Thank you for your helpful comments, according to which we have revised the manuscript.

Summary:

Overall, the manuscript addresses the correlation of AR and P53 with ERβ-positive patients. However, it lacks a sufficient background and rationale in the introduction section.

We have added information regarding TNBC, ERb, p53, and AR to the Introduction.

The results section would benefit from a more detailed description.

We have described the results in detail.

Additionally, the experimental design could be strengthened through a more comprehensive data analysis.

Data have been added according to your suggestion (Table 2, Fig. 5).

Better formatting of both figure legends and references is also recommended.

We have revised both figure legends and the references.

A key limitation of the study is its purely observational nature, analyzing correlations among clinicopathological factors in human patient samples. Therefore, the proposed mechanism link cannot be conclusively established. Detailed suggestions are provided below

We have described the limitations of this study at the end of the Discussion and have deleted ‘KLLN’ in the hypothetical figure (Fig. 6).

Major revisions:

  1. The introduction is overly brief. Elaborating on the functions of AR, ER-β, and p53, as well as providing clear rationales for examining their respective and correlated roles in TNBC, would strengthen the background and improve the overall context. Although some of these rationales are mentioned in the discussion section, I recommend rearranging the content to improve the clarity and flow.

We have elaborated on TNBC, ERb, p53, and AR in the Introduction and rearranged the content.

  1. The results section is insufficiently detailed. Figures or tables are not fully explained in the text.

We have added further detail to the Results, figure legends, and tables.

  1. Figure legends should be properly formatted, including a concise title, a clear description of the staining, scale bars, statistical details, the scoring methods used, etc.

We have revised the figure legends, including the staining, scale bars, statistical details, and scoring methods used.

  1. Representative images of both positive and negative P53, AR and ERβ immunochemistry stainings in Fig.1 will improve the Figure quality and clarity.

We have revised Fig. 1 according to your suggestions.

  1. In Table 1, the author shows that Ki-67 positively correlates with the positivity of ERβ. What is the disease-free interval (DFI) stratified by Ki-67 in relation to ERβ? This will provide a potential prognostic value of ERβ.

We have shown DFI stratified by Ki-67 in relation to ERb; however, no significant result was obtained (Fig.5).

  1. In the results section, lines 72 - 73, a missing figure.

We have provided Table 2 for the relationship between AR and p53.

  1. In the discussion, the authors propose that KLLN may directly promote p53 activity in breast cancer. However, the current evidence based on the correlation between AR and p53 is insufficient to support a link with KLLN. Performing KLLN staining on patient samples could provide additional insight, at least to assess its correlation with p53/AR.

In this study, we have no data regarding KLLN. KLLN is only a hypothetical example to link p53 and AR.  As far as we know, an anti-KLLN antibody that can be used for immunohistochemistry is not commercially available.  We have reduced the description of KLLN in the text and deleted ‘KLLN’ from Fig. 6.

Minor revisions:

  1. Is there any specific reason for choosing postmenopausal women after surgery without adjuvant chemotherapy?

The biological characteristics of TNBC are affected by age. In postmenopausal patients, TNBC is less frequently related to germline mutations in genes such as BRCA than in premenopausal patients. Further, in the previous study, ERb status affected the outcome of TNBC in postmenopausal but not premenopausal patients. We are interested in the heterogeneity of postmenopausal TNBC and conducted an immunohistochemical study using multiple antibodies. For this type of study, surgical material is better than biopsy material regarding the amount of tumor sampled. Patients not receiving adjuvant chemotherapy were chosen in order to determine the baseline characteristics of the tumors. 

  1. H-score can be incorporated to better stratify the staining intensity, as shown in Fig.1 where the intensity of ERβ varies.

We used Allred score (percentage score + intensity score) for ERb; however, the result using the Allred score was almost the same as the present result.

  1. In Table 1, the figure legend indicates both the Pearson chi-squared test and Fisher’s exact test were performed, but only the Pearson chi-squared test was reported in the table.

We have deleted ‘Fisher’s exact test’ in the legend.

  1. Lines 97-100: Missing reference.

We have added references regarding ERb (Refs. 13, 33, and 34).

  1. Lines 140-141: include the Figure number in parentheses.

We have included ‘Table 2’ in parentheses.

  1. The sentence from lines 150 - 155 is overly long, and the transition to discussing mTOR could be smoother.

The sentence has been deleted, as it is unnecessary.

  1. Lines 183-185, citation needed.

We have added citations regarding the immunohistochemical methods (Refs. 18, 23, 43, and 44).

  1. Abbreviations section needs adjustment, for example, HER2.

We have revised the Abbreviations section.

  1. The Reference section needs adjustment, for example, reference 2; unknown characters like O' were included.

We are sorry for the incompleteness of the References, which have been revised carefully.

Reviewer 2 Report

Comments and Suggestions for Authors

This manuscript examined ERβ expression in surgical specimens of TNBC from postmenopausal patients. It was found that ERβ status affected the relationship between the clinical outcome and p53 or androgen receptor status. Collectively, this article should be of broad interest to readers of Int. J. Mol. Sci. and I would recommend the publication of this article after addressing minor corrections.

 minor comments/suggestions

1. Page 2.1) The introduction section is too short. To help readers, please outline the harm of TNBC cancer and common clinical treatment methods. Meanwhile, it will be appropriate to discuss the the research progress of previous studies on the relationship between ERβ in TNBC Associated with p53 and Androgen Receptor. 2) The blockade of immune checkpoints by antagonists has become one of the most promising methods for tumor treatment, especially for TNBC. Besides, peptides based antagonists for cancer immunotherapy is currently among the most promising approaches for the treatment of TNBC. It is appropriate to cite the representative literature which outlines the development of anticancer peptides for cancer immunotherapy (suggest, Chem, 2024, 10, 2390. J. Med. Chem., 2024, 67, 3885).

2. Figure 1.The resolution of Figure 1 is a bit low. Could the author improve the resolution of Figure 1?

3. Please standardize the format of references. For example, some references (eg, reference 1 and 5) lack page numbers.

4. To help readers, please outline the meaning and significance of “disease-free interval”. 

Author Response

To Reviewer 2

Thank you for your favorable and helpful comments, according to which we have revised the manuscript.

 minor comments/suggestions

  1. Page 2.1) The introduction section is too short. To help readers, please outline the harm of TNBC cancer and common clinical treatment methods. Meanwhile, it will be appropriate to discuss the the research progress of previous studies on the relationship between ERβ in TNBC Associated with p53 and Androgen Receptor.

We have outlined the characteristics of TNBC and established treatment methods in the Introduction. We have also introduced studies on the role of ERb, p53, and AR in TNBC, by themselves or in combination.

2) The blockade of immune checkpoints by antagonists has become one of the most promising methods for tumor treatment, especially for TNBC. Besides, peptides based antagonists for cancer immunotherapy is currently among the most promising approaches for the treatment of TNBC. It is appropriate to cite the representative literature which outlines the development of anticancer peptides for cancer immunotherapy (suggest, Chem, 2024, 10, 2390. J. Med. Chem., 2024, 67, 3885).

In the Introduction, we have described treatment options for TNBC, including PARP inhibitors, immune checkpoint inhibitors, and oncolytic peptides, and referred to the suggested article (J. Med. Chem., 2024, 67, 3885, ref); however, the former article (Chem, 2024, 10, 2390) could not be found in PubMed.

2. Figure 1.The resolution of Figure 1 is a bit low. Could the author improve the resolution of Figure 1?

We have revised the Figure 1, showing positive/negative staining for ERb, p53, and AR.

3. Please standardize the format of references. For example, some references (eg, reference 1 and 5) lack page numbers.

We are sorry for the incompleteness of the References in the first draft, which we have revised carefully.

4. To help readers, please outline the meaning and significance of “disease-free interval”. 

The meaning and significance of ‘disease-free interval’ has been described before the clinical outcome results.
